# Expression of Cannabinoid Receptors in the Trigeminal Ganglion of the Horse

**DOI:** 10.3390/ijms242115949

**Published:** 2023-11-03

**Authors:** Rodrigo Zamith Cunha, Alberto Semprini, Giulia Salamanca, Francesca Gobbo, Maria Morini, Kirstie J. Pickles, Veronica Roberts, Roberto Chiocchetti

**Affiliations:** 1Department of Veterinary Medical Sciences, University of Bologna, 37200 Bologna, Italy; rodrigozamithcunha@gmail.com (R.Z.C.); alberto.semprini2@studio.unibo.it (A.S.); giulia.salamanca2@unibo.it (G.S.); francesca.gobbo3@unibo.it (F.G.); maria.morini@unibo.it (M.M.); 2School of Veterinary Medicine and Science, University of Nottingham, Nottingham LE12 5RD, UK; kirstie.pickles@nottingham.ac.uk; 3Bristol Vet School, University of Bristol, Bristol BS40 5DU, UK; veronica.roberts@bristol.ac.uk

**Keywords:** cannabidiol, equine, immunofluorescence, G protein-related receptor 55, peroxisome proliferator-activated receptor gamma, transient receptor potential vanilloid type 1

## Abstract

Cannabinoid receptors are expressed in human and animal trigeminal sensory neurons; however, the expression in the equine trigeminal ganglion is unknown. Ten trigeminal ganglia from five horses were collected post-mortem from an abattoir. The expression of cannabinoid receptors type 1 (CB1R) and type 2 (CB2R), and the cannabinoid-related receptors like transient receptor potential vanilloid type 1 (TRPV1), peroxisome proliferator-activated receptor gamma (PPARɣ), and G protein-related receptor 55 (GPR55) in the trigeminal ganglia (TG) of the horse were studied, using immunofluorescence on cryosections and formalin-fixed paraffin-embedded (FFPE) sections. Neurons and glial cells were identified using fluorescent Nissl staining NeuroTrace^®^ and an antibody directed against the glial marker glial fibrillary acidic protein (GFAP), respectively. Macrophages were identified by means of an antibody directed against the macrophages/microglia marker ionized calcium-binding adapter molecule 1 (IBA1). The protein expression of CB1R, CB2R, TRPV1, and PPARɣ was found in the majority of TG neurons in both cryosections and FFPE sections. The expression of GPR55 immunoreactivity was mainly detectable in FFPE sections, with expression in the majority of sensory neurons. Some receptors were also observed in glial cells (CB2R, TRPV1, PPARγ, and GPR55) and inflammatory cells (PPARγ and GPR55). These results support further investigation of such receptors in disorders of equine trigeminal neuronal excitability.

## 1. Introduction

The most common neuropathic facial pain disorder in horses is trigeminal-mediated (TGM) headshaking, with a reported prevalence of 4% in the United Kingdom equine population [1]. Equine TGM headshaking shares some clinical similarities with human trigeminal neuralgia. However, the underlying pathological mechanisms of TGM headshaking in horses appear to differ from human trigeminal neuralgia, with a functional rather than structural abnormality [2]. Detailed nerve conduction studies of control and headshaking horses identified that the infraorbital branch of the trigeminal nerve in affected horses was sensitised, with a lower threshold for activation than non-affected horses [3].

Research in human trigeminal neuralgia has explored the involvement of the neuromodulatory endocannabinoid system (ECS) [4]. The ECS comprises endocannabinoid molecules involved in signaling processes, along with receptors such as the cannabinoid type 1 (CB1R) and type 2 (CB2R) receptors, and enzymes associated with ligand biosynthesis, activation, and degradation. Cannabinoids derived from Cannabis sativa, including cannabidiol (CBD), cannabigerol, cannabichromene, and cannabinol, as well as synthetic cannabinoids, act on these receptors and other cannabinoid-related receptors such as transient receptor potential (TRP) channels, nuclear peroxisome proliferator-activated receptors (PPARs), and G protein-coupled receptors (GPRs) [5,6].

There is existing evidence suggesting that cannabinoids may effectively alleviate neuropathic pain and hyperalgesia by inhibiting neuronal transmission in pain pathways [7,8]. Although there is a paucity of university-led research on cannabis therapy in equines, in recent years, some interesting studies have shown beneficial therapeutic effects of cannabinoid molecules in horses with allodynia [9], degenerative painful conditions such as osteoarthritis and laminitis [10,11], and also behavioral disturbances [12]. Consequently, considering the pronounced antinociceptive effects of cannabinoids, they could represent a promising therapeutic approach for the clinical management of TGM headshaking, should the appropriate receptors be expressed in the equine trigeminal ganglion. While the localisation of these receptors has been demonstrated in sensory neurons of the equine dorsal root ganglion [13,14], their distribution within the equine trigeminal ganglion is unknown. Therefore, this study aims to immunohistochemically characterise the expression of cannabinoid receptors (CB1R and CB2R) and cannabinoid-related receptors, including TRPV1, PPARɣ, and GPR55, in cryosections and formalin-fixed paraffin-embedded (FFPE) sections of the equine trigeminal ganglion.

## 2. Results

To the authors’ knowledge, the somatotopic organization of the trigeminal nerve and ganglia of the horse has never been investigated; therefore, a portion of the ganglion has been sampled and processed in both cryosections and FFPE. In all sections under examination, there was a voluminous portion of ganglion tissue (with the neuronal component) and a small portion of the adjacent trigeminal nerve (trigeminal root) at the division of the three branches (ophthalmic, maxillary, and mandibular). No differences in immunolabelling were observed across the whole TG examined in all samples. 

The immunoreactivity (IR) of all the studied receptors was visible in both the cryosections and the FFPE sections. In cryosections, all the receptors were widely distributed in neurons, with PPARɣ being the most represented receptor in neurons (95 ± 8%), followed by TRPV1 (90 ± 13%), CB2R (87 ± 18%), and CB1R (80 ± 12%). Since the expression of the GPR55-IR was faint or undetectable in the cryosections of all the subjects analysed, the proportion of TG neurons expressing GPR55-IR was not evaluated; however, the GPR55-IR was more evident in FFPE (see below).

Although the low number of horses from which samples were obtained precludes quantitative evaluation, receptors appeared to have lower expression in older subjects.

Some receptors were also distributed in satellite glial cells (SGCs), vascular cells, and immune/inflammatory cells.

### 2.1. Immunoreactivity in Neurons—Cryosections

Cannabinoid receptor-1 immunofluorescence was faintly expressed by the cell body cytoplasm of 80 ± 12% sensory neurons (440/556 cells counted, *n* = 5) and by large nerve fibres (Figure 1a–c). In some neurons, CB1R-IR was also displayed on the cell membrane (Figure 1d–f). Faint CB1R-IR was also expressed by the nucleoli of sensory neurons.

Cannabinoid receptor-2 immunofluorescence was brightly expressed by the cell body cytoplasm of 87 ± 18% sensory neurons (448/517 cells counted, *n* = 5) and by nerve fibres (Figure 1g–i). Satellite glial cells showed faint CB2R-IR.

Transient receptor potential vanilloid-1 immunofluorescence was brightly expressed by the cell body cytoplasm of 90 ± 13% neurons (496/553 cells counted, *n* = 5) (Figure 2a–c). The TRPV1 immunolabelling was also visible in the growing tip of the axon (axon hillock) and the nerve fibres surrounding the TG sensory neurons (Figure 2d–f). Satellite glial cells showed faint TRPV1-IR.

Peroxisome proliferator-activated receptor gamma immunofluorescence was expressed by the nuclei of the great majority (95 ± 8%) of sensory neurons (509/535 cells counted, *n* = 5) (Figure 2g–i). However, not all the neuronal nuclei showed the same degree of labelling; in fact, in some cells, the nuclei were moderately reactive, while in other neurons, the expression of PPARγ was bright.

G protein-related receptor-55 immunofluorescence was faint or undetectable within the cell body cytoplasm of the TG neurons. For this reason, the count of GPR55 immunolabelled neurons was not possible.

In cryosections, faint CB2-IR was expressed by the cytoplasm of SGCs. Bright PPARγ-IR was expressed by the nuclei of SGCs, as also shown by the co-localisation of the anti-PPARɣ and the anti-GFAP antibodies (Figure 3a–d).

Peroxisome proliferator-activated receptor gamma immunofluorescence was also expressed by the nuclei of interneuronal cells, likely macrophages, as shown by the co-localisation of the anti-PPARγ with the anti-IBA1 antibodies (Figure 3e–h).

Numerous inflammatory/immune cells, which were distributed around and between sensory neurons, showed bright GPR55-IR. However, GPR55-IR was not observed in IBA1-IR cells.

### 2.2. Immunoreactivity in Neurons—FFPE Sections

Cannabinoid receptor 1 immunoreactivity was faintly expressed by the cell body cytoplasm of 63 ± 11% (288/459 cells counted, *n* = 4) sensory neurons, and by the nucleolus of sensory neurons. No CB1R-IR was expressed by nerve fibres (Figure 4a).

Moderate-to-strong CB2R-IR was expressed by the cell body cytoplasm of 72 ± 12% (288/402 cells counted, *n* = 4) sensory neurons and by the nucleus of sensory neurons, as well as by the nuclei of SGCs (Figure 4b) and Schwann cells.

Moderate-to strong TRPV1-IR was observed in the cell body cytoplasm of 81 ± 9% (355/435 cells counted, *n* = 4) of sensory neurons and in the nerve fibres (Figure 4c).

Peroxisome proliferator-activated receptor gamma immunoreactivity with strong intensity was expressed by the neuronal nuclei of 87 ± 6% (284/328 cells counted, *n* = 4) of sensory neurons (Figure 4d,e). It was also expressed by the nuclei of Schwann cells.

G protein-related receptor 55 immunoreactivity was localised in the cell body cytoplasm of 92 ± 8% (304/328 cells counted, *n* = 4) of sensory neurons and by the nuclei of sensory neurons; the immunolabelling was moderate to strong (Figure 4f).

Satellite glial cells showed cytoplasmic CB2R-, TRPV1- and GPR55-IR, and nuclear PPARγ-IR (Figure 4b–f). Notably, not all the SGCs showed PPARγ-IR (Figure 3e).

Small perineuronal and interneuronal immune/inflammatory cells showed a strong cytoplasmic GPR55- and PPARɣ-IR.

Figure 5 shows the graphical representation of the distribution of CB1R, CB2R, TPRV1, GPR55, and PPARɣ in the different cellular elements of the equine TG cells.

Table 1 shows the semiquantitative evaluation of the density of CB1R, CB2R, TRPV1, GPR55, and PPAR-gamma immunoreactivity in different cellular elements of the equine trigeminal ganglia.

## 3. Discussion

Chronic neuropathic pain is characterised by spontaneous pain and hypersensitivity, manifested as hyperalgesia and allodynia, and is often refractory to conventional analgesics and even pharmaceuticals specific for neuropathic pain. Neuropathic pain therefore, most of the time, represents an unmet therapeutic need. Trigeminal-mediated headshaking is likely to be the most common neuropathic facial condition causing pain in horses [1]. It can be a significant source of pain, compromised welfare, and wastage in horses, alongside safety concerns for riders and handlers. Therefore, elucidating better, more rational treatments to alleviate this condition is of paramount importance. The ECS plays a major role in the pain trafficking pathway, at both central and peripheral levels, and modification of this system may open novel therapeutic targets [15,16]. To date, it is known that cannabis-based drugs have important therapeutic potential in people, not only in treating inflammatory diseases (such as osteoarthritis and inflammatory bowel disease), in dermatological diseases such as atopic dermatitis and psoriasis, but also in neurodegenerative diseases such as Alzheimer’s, multiple sclerosis and Parkinson’s, and neurological pathologies that induce seizures and eating disorders [17,18,19]. Cannabinoid-based drugs may also be useful in human trigeminal neuralgia [7,20,21]. Recently, Longworth et al. [21] reported that cannabis-based medicines offered therapeutic benefits in 6 out of 8 treatments for people with chronic facial pain, and in all cases, they did not cause undesirable effects.

While there are studies reporting the expression of cannabinoid receptors, endocannabinoids, and their related enzymes in the TG of rodents, and the therapeutic efficacy of CBD in modulating nociceptive transmission and head pain [16,22,23], research on the ECS in the nervous system of horses is in its infancy. Although interesting electrophysiological studies on trigeminal nerve conduction in horses are available [3,24], no functional studies related to the influence of the ECS on the mechanisms of neuropathic pain have been published. In rodents, endocannabinoids may inhibit the trigeminal release of neuropetides that control nociceptive inputs through a CB1R-dependent mechanism [25]. Since the expression of neuropetides regulating the nociceptive pathways has been also described in horse sensory neurons [26], it is plausible to hypothesize that the ECS might act with a similar mechanism in this species.

The evidence that cannabinoid receptors (CB1 and CB2) and cannabinoid-related receptors (TRPV1, PPARɣ, and GPR55) are extensively expressed in the majority of equine trigeminal ganglion neurons is indeed an encouraging discovery. In the present study, two different methodological procedures were applied to detect the cannabinoid and cannabinoid-related receptors in the trigeminal ganglia, i.e., immunofluorescence on cryosections and immunohistochemistry on FFPE sections; as these receptors are molecules that could be degraded by the processes used for paraffin embedding. The fact that the antibodies were able to identify neurons in paraffin-fixed tissues is an encouraging finding, which will allow for future comparisons of labelling with TG obtained from horses with TGM headshaking, where tissues were collected and processed using standard techniques for paraffin embedding.

### Immunoreactivity in Trigeminal Neurons

Cannabinoid receptor-1 is widely expressed in the nervous pathways of the nociceptive system of rodents [22,23,27,28,29], and in other mammals, including dogs [30] and horses [13]. While in rats, CB1R is expressed almost exclusively by myelinated neurons in both the TG [22] and DRG [28], in horses the receptor seems to be localised in neurons of different dimensions.

Cannabinoid receptor-1 can be activated by endogenous and exogenous cannabinoids, and it can modulate the release of nociceptive neurotransmitters. Several studies have shown that CB1 receptor activation by endocannabinoids causes depression of neuronal excitability and neurotransmitter release in presynaptic primary neurons [31,32]. However, there is contradictory evidence showing that endocannabinoids may also contribute to the potentiation of neurotransmission through CB1 receptors, as shown by Zhang et al. [33], who found that the endocannabinoid anandamide (AEA) may cause elevation of the intracellular Ca2+ concentration in the small neurons of the rat TG. Another study, which was undertaken to elucidate the effects of certain cannabinoids on nociceptive TG cultured neurons, showed that the predominant effects of AEA on TG nociceptors were excitatory, and in part, also mediated by TRPV1 [34]. Alternatively, Akerman et al. [35] showed that the cannabinoid receptor agonist WIN55,212, which acts on both CB1 and CB2 receptors, was able to significantly inhibit the activation of trigeminal neurons by electrical stimulation of the dura mater by acting and inhibiting neurons with A-fiber and C-fiber inputs. All of these findings highlight the necessity for further study of the ECS as it relates to nociception, with particular attention to the role of CB1R in sensory neurons.

Cannabinoid receptor-2 was initially considered a ‘peripheral cannabinoid receptor’, since it is abundantly expressed in the immune system, including in horses [14]. However, a growing body of evidence indicates that CB2R is not only expressed in brain microglia during neuroinflammation [36], but is also expressed in peripheral sensory neurons, as shown in humans [37], rats [23,30], and horses [13]. Cannabinoid receptor-2 shows vast therapeutic potential. The most promising indications include neuropathic pain, in which CB2R agonists have proven helpful [38,39]. Therefore, the finding that a large proportion of equine TG neurons express CB2R (as seen in the current study) reinforces the evidence that CB2R may play a great role in pain modulation in horses, as also proposed by Sánchez-Aparicio et al. [10].

In both a former study [13] and the current one, CB2R-IR has also been expressed by SGCs; this evidence is consistent with that described in rat sensory neurons [40].

The TRP vanilloid 1 (TRPV1) ion channel is usually expressed by DRG nociceptors of mammals [30,41,42,43], including horses [14]. In rats, TRPV1 receptors are known to be present in trigeminal neurons with C-fibre inputs [44]. It has been suggested that the role of TRPV1 in neuropathic pain may include sensitisation of intact peripheral terminals, ectopic activity in injured axons (through the activation of axonal TRPV1 receptors), or a contribution to the modulation of transmitter release [45].

It has been shown that TRPV1 antagonists reduced pain sensitivity in models of neuropathic pain [46] and provided pain relief in models of inflammatory pain [47]. In addition, the desensitisation of TRPV1 located in DRG and TG has exhibited analgesic effects, making it a potential therapeutic target for the treatment of neuropathic pain [48].

A recent study has suggested a role for TRPV1 channels in the modulation of synaptic transmission through activation by endocannabinoids, such as anandamide [49]. Additionally, endocannabinoid-like molecules and endovanilloids can desensitise TRPV1, as well as phytocannabinoids such as CBD [50], which shows anti-nociceptive, analgesic, and anti-inflammatory effects [51].

Peroxisome Proliferator Activated Receptor ɣ, like the other PPAR receptors, acts as a transcription factor, modulating various physiological functions, including lipid metabolism. It is principally expressed in fatty and vascular tissue [52]; however, it has also been localised in brain tissue, where its activation reduced neurodegeneration [53]. There is some evidence indicating that PPARγ plays a beneficial role in various neurological diseases and that its activation may represent a potential target for the treatment of numerous acute and chronic neurological diseases [54]. Indeed, the activation of PPARγ has reduced neuroinflammation in neuropathic pain [55,56]. The role of PPARγ during inflammatory compression of the trigeminal nerve was investigated by Lyons et al. [57], who demonstrated its important role in pain sensitivity and alleviation of allodynia with PPARγ agonists. A further study suggests that drugs targeting spinal PPARγ may yield important therapeutic effects for neuropathic pain [58]. Cannabinoids produce analgesic effects by acting on multiple pain targets in the peripheral and central nervous systems [59]. Cannabidiol, particularly CBD, due to its interaction with PPARγ, may be useful for treating neuropathic pain [60,61].

In studies performed on equine spinal ganglia, immunoreactivity for the PPARγ receptor was expressed by sensory neurons, SGC, and other unidentified interneuronal cells [13,14]. The data reported in the current study on horse TG are consistent with those expressed by the spinal ganglia. The localisation of PPARɣ-IR in SGCs of the horse TG may play an important role in ganglion functional homeostasis, as also indicated in mice by Chu et al. [62]. Also, the expression of PPARɣ-IR in the macrophages of the horse TG seems to be an interesting finding, as there are observations suggesting that PPARγ signaling in macrophages may be a potential therapeutic target for the treatment of acute pain development [63,64].

G protein-coupled receptor 55 (GPR55), which researchers have proposed classifying as a third cannabinoid receptor [65,66], has been identified in the sensory neurons of different species, including dogs, rats [30], and horses [14], and in canine inflammatory cells [67]. While in cryosections, the GPR55 signal was so faint that it prevented cell counting; in FFPE sections, GPR55-IR was strongly expressed (see below).

Unexpectedly, GPR55-IR was not observed in IBA1-IR cells, as previously shown in the horse DRG [14]. This evidence indicates that macrophages (or dendritic or microglia cells) of the equine TG do not express GPR55-IR. However, GPR55-IR has been observed in unidentified inflammatory/immune cells in both cryosections and FFPE sections of the TG, suggesting an active role of the receptor in TG homeostasis and immunity.

In a recent study in mice, GPR55 signaling contributed to the induction of inflammatory responses and chronic pain via the recruitment of neutrophils, monocytes/macrophages, and T-cells, possibly providing a new target for reducing pain [68]. Another study reported that GPR55-KO mice did not develop mechanical pain in chronic pain models developed by peripheral nerve injury and peripheral tissue inflammation [69]. Therefore, molecules that block GPR55 (such as CBD) could potentially help patients with diverse ailments, including chronic pain.

Immunoreactivity in satellite glial cells—localisation of some receptors in SGCs (PPARγ, CB2R, GPR55) and inflammatory/immune cells (PPARγ and GPR55) seems to be relevant, because a better understanding of the molecular crosstalk between TG neurons and their surrounding cells may aid in the identification of novel targets for the development of more effective analgesics [70]. In response to injury signals, SGCs and macrophages accumulate around primary sensory neurons, forming a macrophage-SGC-neuron triad, in which macrophages and SGCs work together to enhance and prolong neuropathic pain through ATP and other inflammatory mediators, which promote the development of inflammation-related neuropathic pain [70].

Satellite glial cells that envelop the cell bodies of primary sensory neurons in DRG and TG support normal sensory transmission and nociception by maintaining metabolic and ionic homeostasis [40]. In neuropathic pain and inflammation models, SGCs proliferate and become activated [71,72], resulting in an increased expression of various molecules, such as proinflammatory cytokines [73,74,75] and neurotrophins [76,77,78], and changes in functional gap junctions [79,80]. The mechanisms by which peripheral nerve injury causes activation of SGCs may involve ATP, with ATP released from damaged neurons believed to be one of the critical mediators involved in activating SGCs through the stimulation of purinergic receptors (as reviewed by Ohara et al. [77]). Recent studies have demonstrated that SGCs are able to modulate neuronal excitability, leading to neuropathic pain [76,78].

In the TG, satellite glial cells have been recognised as significant contributors to the regulation of neuronal function in various physiological and pathological conditions, including pulpitis, inflammation of the temporomandibular joint, headaches, and other orofacial neuropathic or inflammatory pain [81,82]. Cannabinoids may have a therapeutic role in neuropathic pain, also reducing the synergy between neurons and SGCs in the nociceptive pathways.

The small sample size, potential regional differences, and lack of close age matching in our cases could represent a significant limitation, as well as the limited number of horses and the uncertainty regarding their health status. Further studies involving a larger number of subjects are required to confirm our results. It cannot be ruled out that some factors, such as the unknown underlying pathological conditions of the horses in the study or the medications they received, could potentially alter the CB1R, CB2R, TRPV1, GPR55, and PPARɣ expression in tissues. 

## 4. Materials and Methods

### 4.1. Animals

Ten whole trigeminal ganglia (TG) from both sides (right and left) of five healthy horses, which were slaughtered for consumption, were harvested post-mortem from the head. The age of the horses ranged from 10 months to 22 years (6 ± 5 years; mean ± standard deviation), and their weight ranged from 67 kg to 374 kg (274 ± 121 kg). The breeds included 2 Italian thoroughbred, 2 half-breeds and 1 French saddle horse.

The clinical history of the horses was unknown, but all had been assessed as fit for slaughter for human consumption upon ante-mortem inspection. Heads were routinely removed during carcass preparation and were cut off along the sagittal plane of the head. This cut revealed the trigeminal roots, the attached trigeminal ganglia, and portions of the ophthalmic, maxillary, and mandibular nerves (approximately 1.5 cm in length), which were quickly removed and immersed in the fixative. Dissection was performed within 0.5–1 h of slaughter. The inclusion criteria for the selection of the animals were the normal gross and histological TG appearance.

### 4.2. Immunofluorescence on Cryosections

Five right TG from the five horses were fixed for 48 h at 4 °C in 4% paraformaldehyde in phosphate buffer (0.1 M, pH 7.2). They were subsequently rinsed in phosphate-buffered saline (PBS; 0.15 M NaCl in 0.01 M sodium phosphate buffer, pH 7.2) and stored at 4 °C in PBS containing 30% sucrose and sodium azide (0.1%). The following day, the tissues were transferred to a mixture of PBS-30% sucrose–azide and Optimal Cutting Temperature (OCT) compound (Sakura Finetek Europe, Alphen aan den Rijn, The Netherlands) at a ratio of 1:1 for an additional 24 h before being embedded in 100% OCT in Cryomold^®^ (Sakura Finetek Europe, Alphen aan den Rijn, The Netherlands). The sections were prepared by freezing the tissues in isopentane cooled in liquid nitrogen. Cryosections (14 μm thick) (*n* = 60) of trigeminal ganglia were cut on a cryostat and mounted on polylysinated slides (*n* = 22).

The cryosections were hydrated in PBS and processed for immunostaining. To block non-specific binding, the sections were incubated in a solution containing 20% normal donkey serum (Colorado Serum Co., Denver, CO, USA), 0.5% Triton X-100 (Sigma Aldrich, Milan, Italy, Europe), and 1% bovine serum albumin in PBS for 1 h at room temperature (RT) (22–25 °C). Serial cryosections of the same TG in all five cases were incubated in a humid chamber overnight at RT with different antibodies directed against CB1R, CB2R, TRPV1, GPR55, and PPARɣ (single immunostaining) or with a cocktail of primary antibodies (double immunostaining) (Table 2 and Table 3), diluted in 1.8% NaCl in 0.01 M PBS containing 0.1% sodium azide.

Primary antibody Suppliers: Abcam, Cambridge, UK; Alomone, Jerusalem, Israel; Dako Cytomation, Golstrup, Denmark; Novus Biologicals, Littleton, CO, USA; Santa Cruz Biotechnology, Santa Cruz, CA, USA; Thermo Fisher Scientific, Waltham, MA, USA. Abbreviations: IF, immunofluorescence; IHC, immunohistochemistry.

Secondary antibody Suppliers: Abcam, Cambridge, UK; Dako, Glostrup, Denmark; Jackson Immuno Research Laboratories, Inc Baltimore Pike, PA, USA. Thermo Fisher Scientific, Waltham, MA, USA. Vector Laboratories, Burlingame, CA, USA.

Table 3 summarizes the combination of primary and secondary antibodies used in double labelling experiments.

After washing in PBS (3 × 10 min), the sections were incubated for 1 h at RT in a humid chamber with the secondary antibodies (Table 4) diluted in PBS.

The sections were then washed in PBS (3 × 10 min) and counterstained with a blue fluorescent Nissl stain solution (NeuroTrace^®^, # N-21479, dilution 1:200) (Molecular Probes, Eugene, OR, USA), which was used to label the TG neurons and the satellite glial cells (SGCs) and to determine the percentage of neurons that immunoreacted to each of the markers.

To identify the SGCs, the antibody anti-glial acidic fibrillary protein (GFAP) was also utilised. In addition, since cannabinoid and cannabinoid-related receptors might also be located on inflammatory cells, the anti-ionised calcium binding adapter molecule 1 antibody (IBA1), which recognises the microglia in the central nervous system (CNS) and the macrophages outside the CNS, was utilised. The cryosections were then washed in PBS (3 × 10 min) and mounted in buffered glycerol at pH 8.6.

### 4.3. Immunohistochemistry on FFPE

Five left TG were dissected from five horses (in one case, the sampled tissue was insufficient for further analysis), fixed in 10% (*vol*/*vol*) phosphate-buffered formalin, and paraffin-embedded according to standard procedures. Subsequently, three-micron-thick consecutive sections were cut and processed for immunohistochemistry (*n* = 20). The sections from each sample were dewaxed and rehydrated. Endogenous peroxidase was blocked by immersion in 3% H_2_O_2_ in methanol for 30 min at RT. The dilutions of the primary antibodies are reported in Table 2. Antigen retrieval (citrate pH6; 10 min; MW: 750 W) was followed by cooling at RT for 20 min. Blocking of non-specific antigenic sites was achieved by incubating the slides in a solution of 10% normal goat serum in PBS (blocking solution) for 30 min at RT, and incubated overnight in a humid chamber at 4 °C with the primary antibodies diluted in the blocking solution afterwards.

The slides were rinsed in TRIS buffer and then incubated with a secondary anti-rabbit antibody (biotinylated goat anti-rabbit immunoglobulins; Dako, Glostrup, Denmark) diluted to 1:200 in the blocking solution. After two washes in TRIS buffer, immunoreactions were detected with avidin-biotin immunoperoxidase (Vectastain Elite ABC Kit, Vector Laboratories, Burlingame, CA, USA) and visualised with the chromogen 3,3′-diaminobenzidine (0.05% *w*/*v*, cat# ACB999, Histo-Line Laboratories, Pantigliate, Milan, Italy). The slides were counterstained with Harris haematoxylin (cat# 01HEMH2; Histo-Line Laboratories) and permanently mounted with DPX medium (Fluka, Riedel-de Haen, Germany). Images were acquired with an optical microscope (Eclipse E600; Nikon, Shinjuku, Japan) equipped with the Imaging Source “33” Series USB 3.0 Camera (cat# DFK 33UX264; Bremen, Germany).

### 4.4. Quantitative Analysis of the Immunoreactivity

The immunoreactivity of the antibodies was evaluated, and its cellular localisation (membranous, cytoplasmic, and/or nuclear) was reported. The intensity of the expression was also evaluated as faint, moderate, and bright (immunofluorescence)/strong (immunohistochemistry) in images acquired using the same exposure times (for immunofluorescence in cryosections), and through microscopy observations (for immunohistochemistry on paraffin-embedded sections).

A minimum of one hundred Nissl-stained neurons were counted for each cannabinoid and cannabinoid-related receptor in cryosections, following the protocol published by Chiocchetti et al. [13].

In the FFPE sections, immunopositive neurons were counted in a total of 5 fields at 20× magnification under the optical microscope and compared to the total number of cells present in the same cell compartment.

The relative percentages of immunopositive neurons were expressed as Mean ± standard deviation (SD).

### 4.5. Specificity of the Primary Antibodies

The rabbit anti-CB1R [13], -CB2R [83], -TRPV1 [14,30], and -PPARɣ [14] antibodies had previously been tested on horse tissues using Western blot (Wb) analysis. The immunogen used to obtain the anti-GPR55 antibody was a synthetic 20 amino acid peptide from the third cytoplasmic domain of Human GPR55 in amino acids 200–250. The homology between the full amino acid sequences of the horse and human GPR55 was 80%, and the correspondence with the specific sequence of the immunogen was 78% (https://www.uniprot.org/, accessed on 7 January 2018). This antibody, which has recently been used in horse sensory neurons [14], had previously been tested on rat and dog dorsal root ganglia (DRG) using immunohistochemistry [30] and on mice tissues using Wb analysis [84]. However, the Wb analysis had not been carried out on horse tissue.

Marker for macrophages—the goat anti-IBA1 antibody, recently used on horse tissue [12,14], was directed against a peptide with the sequence C-TGPPAKKAISELP, from the C Terminus of the porcine IBA1 sequence. Horse and porcine IBA1 molecules share 92.3% identity (https://www.uniprot.org/, accessed on 30 June 2019), and it is plausible that the antibody used can also recognise IBA1 in the horse.

### 4.6. Specificity of the Secondary Antibody

The specificity of the secondary antibodies was tested by applying them to the sections after omitting the primary antibodies. No immunolabelled cells were detected after omitting the primary antibodies.

### 4.7. Fluorescence Microscopy

The preparations were examined, by the same observer on a Nikon Eclipse Ni microscope (Nikon Instruments Europe BV, Amsterdam, The Netherlands, Europe) equipped with the appropriate filter cubes. The images were recorded with a DS-Qi1Nc digital camera and NIS Elements software BR 4.20.01 (Mountain View, Ottawa, ON, Canada). Slight contrast and brightness adjustments were made using Corel Photo Paint, whereas the figure panels were prepared using Corel Draw (Mountain View).

## 5. Conclusions

In conclusion, this study provides significant insights into the expression of cannabinoid receptors (CB1 and CB2) and cannabinoid-related receptors (GPR55, PPARγ, and TRPV1) in the TG neurons and SGCs of horses. To the authors’ knowledge, this is the first study to demonstrate this expression. The positive findings demonstrate the presence and potential functional significance of these receptors in the equine TG, highlighting their potential role in the modulation of trigeminal nerve function and neuropathic pain pathways. Considering the pronounced antinociceptive effects of cannabinoids and the expression of their receptors in the TG, these findings hold promise for the therapeutic application of cannabinoids in managing headshaking in horses. By targeting the endocannabinoid system, modulation of the trigeminal neural network and subsequent alleviation of trigeminal-mediated headshaking may be achievable. Investigation of receptor expression in TGM headshaking, and whether it differs from unaffected horses, is warranted and now being undertaken. Such research endeavors could ultimately lead to the development of novel cannabinoid-based therapies for the clinical management of this debilitating condition, thereby enhancing the well-being, performance, and quality of life of these horses and improving safety for riders and handlers.

## Figures and Tables

**Figure 1 ijms-24-15949-f001:**
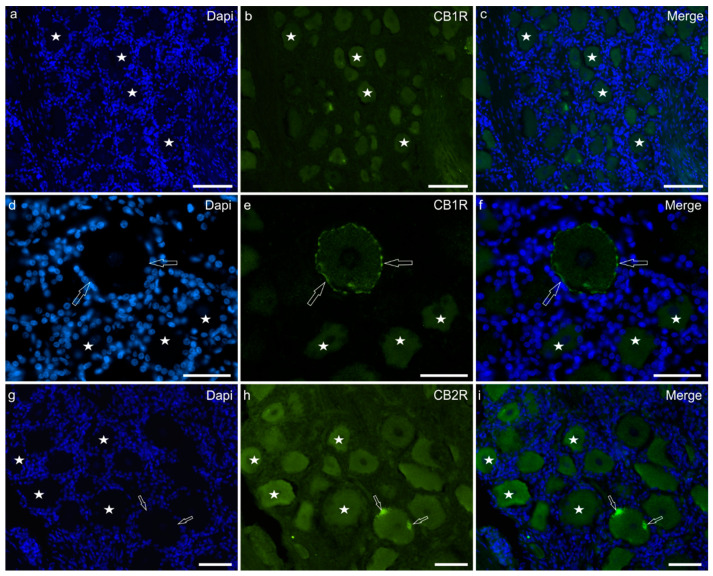
Photomicrographs of equine trigeminal ganglion cryosections showing immunoreactivity for the cannabinoid receptor type 1 (CB1R) (**a**–**f**) and type 2 (CB2R) (**g**–**i**). (**a**–**f**) Stars indicate a few sensory neurons showing faint cytoplasmic CB1R immunoreactivity of the cell body cytoplasm. The open arrows indicate a neuron in which CB1R immunoreactivity was also expressed by the cell membrane. (**g**–**i**) Stars indicate neurons expressing moderate CB2R immunoreactivity. Two open arrows indicate autofluorescent pigments which were confined to the edges of the cell. Bar: 50 µm.

**Figure 2 ijms-24-15949-f002:**
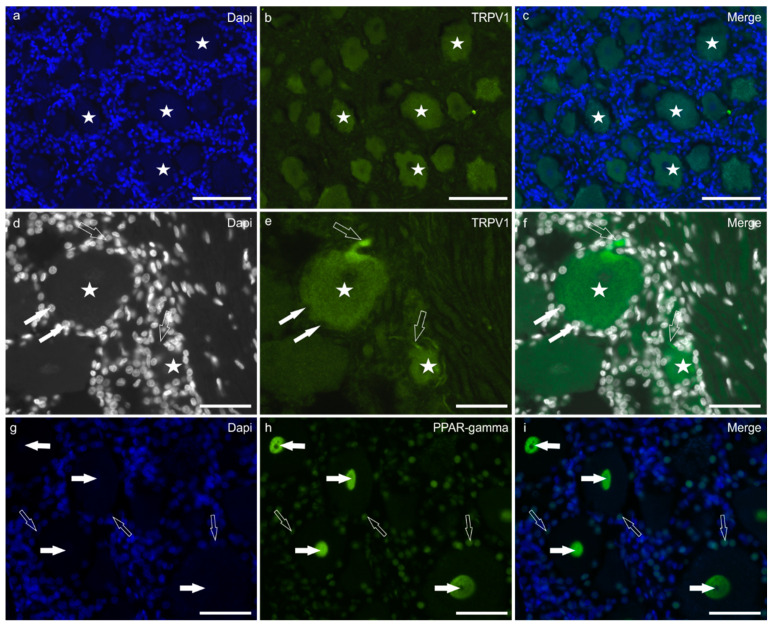
Photomicrographs of equine trigeminal ganglion cryosections showing immunoreactivity for the cannabinoid-related receptor TRPV1 (transient receptor potential vanilloid 1) (**a**–**f**), and PPARγ (Peroxisome proliferator-activated receptor gamma) (**g**–**i**). (**a**–**f**) Stars indicate sensory neurons expressing moderate-to-bright TRPV1 immunoreactivity. The white arrows indicate the Dapi labelled nuclei of satellite glial cells surrounding a large neuron expressing faint TRPV1 immunoreactivity. The open arrows indicate two TRPV1 immunoreactive axon hillocks arising from two sensory neurons. (**g**–**i**) The white arrows indicate neuronal nuclei expressing bright PPARγ immunoreactivity. The open arrows indicate the nuclei of satellite glial cells showing moderate PPARγ immunoreactivity. Bar: 50 µm.

**Figure 3 ijms-24-15949-f003:**
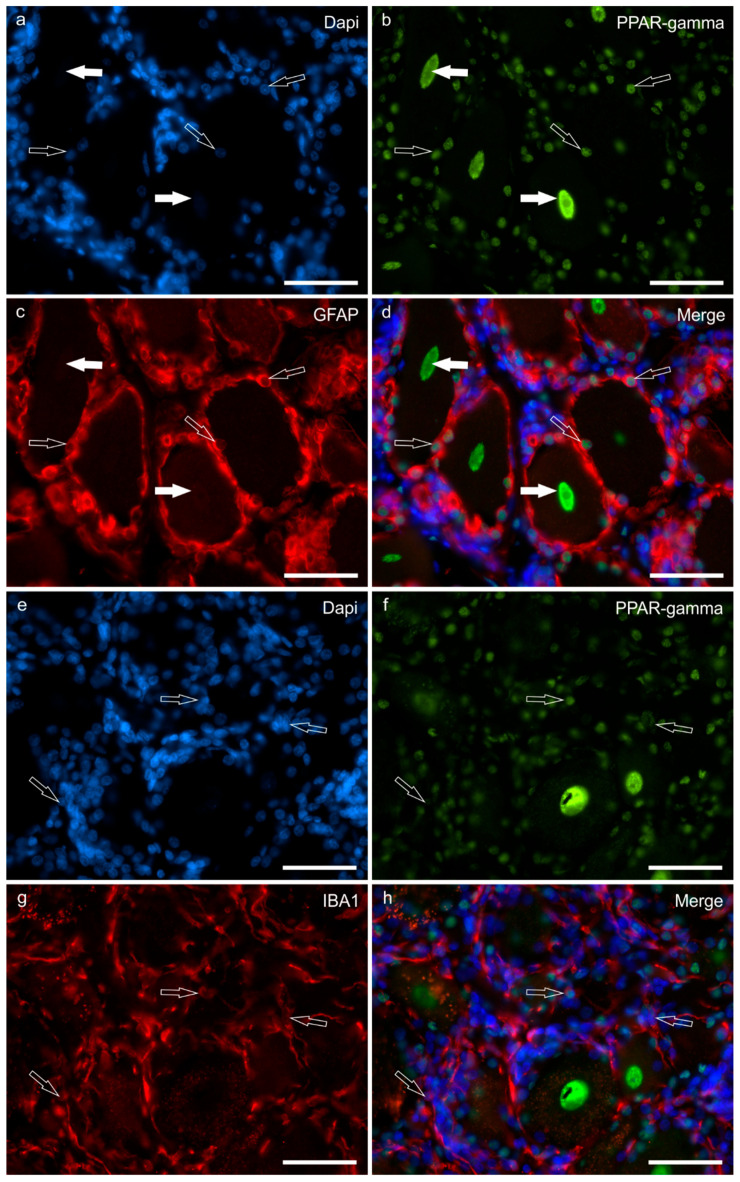
Photomicrographs of equine trigeminal ganglion cryosections showing the co-localisation between the antibody anti-PPARγ (Peroxisome proliferator-activated receptor gamma) and the satellite glial cells marker GFAP (glial acidic fibrillary protein), (**a**–**d**) and between the antibody anti-PPARγ and the macrophage marker IBA1 (**e**–**h**). (**a**–**d**) The white arrows indicate two neuronal nuclei showing bright PPARγ immunoreactivity; the open arrows indicate the dapi labelled nuclei of GFAP immunoreactive satellite glial cells showing moderate PPARγ immunoreactivity. (**e**–**h**) The open arrows indicate the dapi labelled nuclei of IBA1 immunoreactive macrophages which co-expressed PPARγ immunoreactivity. Bar: 50 µm.

**Figure 4 ijms-24-15949-f004:**
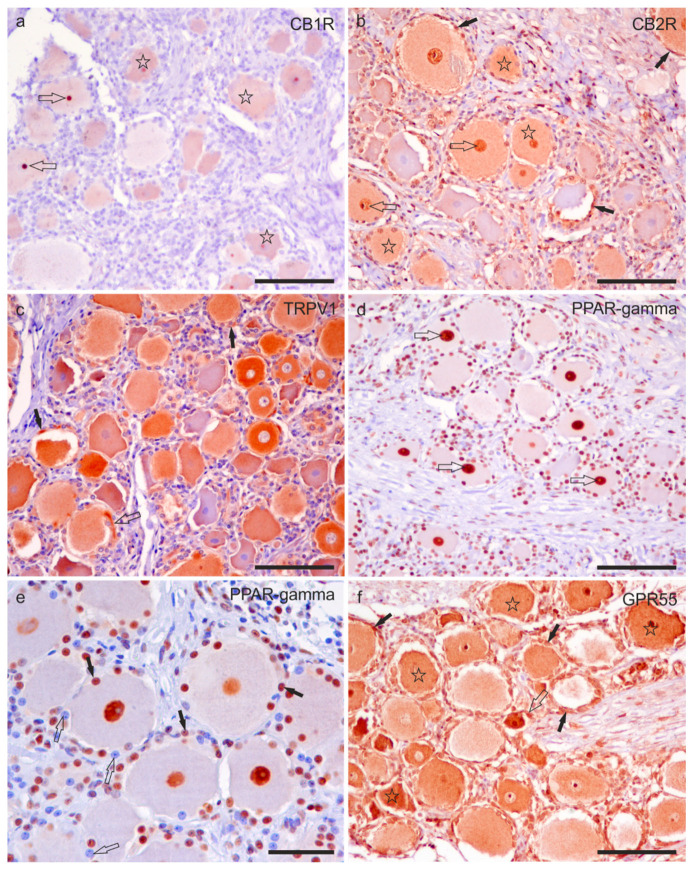
Photomicrographs of formalin-fixed paraffin embedded sections of equine trigeminal ganglion showing immunoreactivity for the cannabinoid receptor type 1 (CB1R) (**a**) and type 2 (CB2R) (**b**), transient receptor potential vanilloid 1 (TRPV1) (**c**), perossisome proliferator-activated receptor gamma (PPARγ) (**d**,**e**), and G-protein coupled receptor 55 (GPR55) (**f**). (**a**) Stars indicate sensory neurons showing faint CB1R immunoreactivity. The open arrows indicate two strongly immunoreactive nucleoli for the CB1R. (**b**) Stars indicate sensory neurons which showed moderate CB2R immunoreactivity. The open arrows indicate the neuronal nuclei expressing strong CB2R immunoreactivity. Black arrows show satellite glial cells surrounding the sensory neurons, showing faint-to-moderate CB2R immunoreactivity. (**c**) A large proportion of sensory neurons expressed moderate-to-strong TRPV1 immunoreactivity. Axon hillocks (open arrow) were also positive for TRPV1, as well as satellite glial cells (black arrows). (**d**) The open arrows indicate the nuclei of the sensory neurons, showing strong PPARγ immunoreactivity. (**e**) Not all nuclei of satellite glial cells showed PPARγ immunoreactivity; some were strongly reactive (black arrows), whereas some appeared negative (open arrows). (**f**) Stars indicate the cell body cytoplasm of sensory neurons with moderate-to-strong GPR55 immunoreactivity. The open arrows indicate the axon hillock of a small neuron which expressed strong GPR55 immunoreactivity. The black arrows indicate some satellite glial cells which were positive for GPR55. Bars: (**a**–**d**,**f**): 100 µm; (**e**): 50 µm.

**Figure 5 ijms-24-15949-f005:**
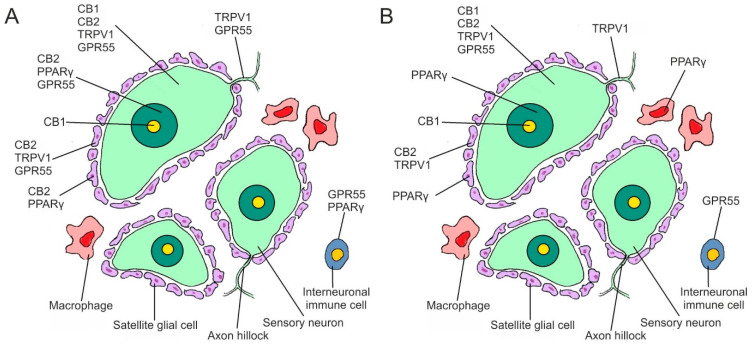
Graphical representation of the distribution of the cannabinoid receptors 1 (CB1R) and 2 (CB2R), and the cannabinoid-related receptors transient receptor potential vanilloid 1 (TRPV1), G protein-coupled receptor 55 (GPR55), and nuclear peroxisome proliferator-activated receptor gamma (PPARγ), in the different cellular elements of the equine trigeminal ganglion. (**A**): Expression of immunofluorescence in cryosections; (**B**): Expression of immunohistochemistry on formalin-fixed paraffin-embedded sections.

**Table 1 ijms-24-15949-t001:** Semiquantitative evaluation of the density of CB1R, CB2R, TRPV1, GPR55, and PPAR-gamma immunoreactivity in different cellular elements of the equine trigeminal ganglia. The immunoreactive cells were graded as: −, negative; +, faint; ++, moderate; +++, bright. Abbreviations: Cryo, cryosections; FFPE, formalin fixed paraffin embedded sections, IF, immunofluorescence; IHC, immunohistochemistry.

Receptors	CB1R	CB2R	TRPV1	GPR55	PPAR-ƴ
Neurons	+ IF Cryo + IHC iFFPE	++ IF Cryo ++ IHC FFPE	+++ IF Cryo +++ IHC FFPE	+/− IF Cryo ++/+++ IHC FFPE	+++ IF Cryo +++ IHC FFPE
Satellite glial cells	− IF Cryo − IHC FFPE	++ IF Cryo +++ IHC FFPE	− IF Cryo − IHC FFPE	− IF Cryo ++ IHC FFPE	+++ IF Cryo +++ IHC FFPE
Inflammatory or immune cells	− IF Cryo − IHC FFPE	− IF Cryo − IHC FFPE	− IF Cryo − IHC FFPE	− IF Cryo +++ IHC FFPE	+++ IF Cryo +++ IHC FFPE

**Table 2 ijms-24-15949-t002:** Primary antibodies used in the study.

Primary Antibody	Host	Code	Dilution	Source
CB1R	Rabbit	ab23703	1:100 (IF); 1:400 (IHC)	Abcam
CB2R	Rabbit	PA1-744	1:250 (IF); 1:200 (IHC)	Thermo Fisher
GPR55	Rabbit	NB110-55498	1:200 (IF); 1:200 (IHC)	Novus Biol.
GFAP	Chicken	Ab4674	1:800 (IF)	Abcam
IBA1	Goat	NB100-1028	1:80 (IF)	Novus Biol.
PPARɣ	Rabbit	Ab45036	1:300 (IF); 1:400 (IHC)	Abcam
TRPV1	Rabbit	ACC-030	1:200 (IF); 1:300 (IHC)	Alomone

**Table 3 ijms-24-15949-t003:** Details of the primary and secondary reagent combination in the double labelling experiments.

Double Labelling Experiment	Secondary Reagent Combination
Rabbit anti-PPARγ	Donkey anti-Rabbit 488
Chicken anti-GFAP	Donkey anti-Chicken TRITC
Rabbit anti-PPARγ	Donkey anti-Rabbit 488
Goat anti-IBA1	Donkey anti-Goat 594
Rabbit anti-GPR55	Donkey anti-Rabbit 488
Chicken anti-GFAP	Donkey anti-Chicken TRITC
Rabbit anti-GPR55	Donkey anti-Rabbit 488
Goat anti-IBA1	Donkey anti-Goat 594

**Table 4 ijms-24-15949-t004:** Secondary antibodies used in the study.

Secondary Antibody	Host	Code	Dilution	Source
Anti-chicken TRITC	Donkey	703-025-155	1:200	Jackson
Anti-goat 594	Donkey	ab150132	1:600	Abcam
Anti-rabbit 488	Donkey	A-21206	1:1000	Thermo Fisher
Anti-rabbit biotinylated	Goat	BA-1000	1:200	Vector laboratories

## Data Availability

The data that support the findings of this study are available from the corresponding author upon reasonable request.

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
