# Peer review of "Expression of Cannabinoid Receptors in the Trigeminal Ganglion of the Horse"

_ijms, 2023, doi:10.3390/ijms242115949_

Round 1

Reviewer 1 Report

Comments and Suggestions for Authors

In this descriptive study, the authors looked at cannabinoid and cannabinoid-related receptors in the horse trigeminal ganglia of 5 horses. They found them.

The introduction is short but given the aims of this descriptive study, they are adequate but could be improved with more depth on the functional role of cannabinoids and the use of cannabinoids in equids.

The materials and methods need considerably more detail before they can be suitably evaluated. It is not clear where in each ganglia staining took place or how many slices were obtained. Each ganglion was apparently used for single and double staining, which would suggest different sections/slides were used for each. This should be explained. Was the order standardized or randomized?

Based on the description it reads as if primary ABs from the same host were stained for at the same time in the multiple tag steps. How were these differentiated when they were subsequently stained by the same secondaries? The specific staining rounds are confusing and may benefit from a table. That may allow an opportunity to also illustrate where in each ganglion each staining took place.

The methods surrounding cell counting should also be better explained.  It is not clear why the numbers differ between proteins (e.g. 517, 556, 535, 435, 459, etc.). From the current description, it reads as if this represents 100+ cells from each animal (why that number is not 500 is not clear). Treating each as a separate value is a statistical error. Each animal is an n=1, therefore the average cell number (e.g. out of 100) positive for a protein should be reported as a percentage for each animal (82/100, 75/100, 90/100, etc.) rather than grouped as independent events out of ~500.

The methods indicate that the FFPE was conducted in 5 ganglia however the results have an n=4. Please clarify the discrepancy.

The study involved a very wide range of ages (10 months  – 22 years) and animal sizes (67kg -374 kg). with such disparity in the sample size, the authors should comment on any trends or differences based on age and size in the results. In addition, a comment should be added to further express the degree of animal variation as an additional study limitation.  

Intext references are needed to several statements without support inclusive of the 1st line of 4.5 where wb analysis in horses is mentioned. Also how/where the homology of proteins was determined.

References are also needed for claims in the discussion, for example in the 4th line where trigeminal mediated headshaking is mentioned as the most common neuropathic pain facial condition of horses.

In general, statements that cannot be substantiated should be omitted or revised.

It is unclear why contrast and brightness adjustments were needed. Adjusting these parameters can result in misinterpreting autofluorescence.

Was Nissl staining included as part of the secondary validation (section 4.6)? It reads as if no cells were identified when in fact they should still be identified even if receptors were not tagged.  

Please increase the font size of Figure 5 or break it up into 2 legible figures.

Unfortunately, references can reflect on the quality of the science. Even the use of software should be double-checked before submitting to a journal. Many inconsistencies exist in how references are cited including the use of capital letters and abbreviations for journals. For examples see refs 11, 14, 43. Please double-check references for consistency, accuracy, and journal compliance before submitting

Author Response

Reviewer # 1

In this descriptive study, the authors looked at cannabinoid and cannabinoid-related receptors in the horse trigeminal ganglia of 5 horses. They found them.

Response: We thank the Reviewer very much for his/her careful review and valuable suggestions, which we have tried to accommodate and resolve in the text of our paper.

  1. The introduction is short but given the aims of this descriptive study, they are adequate but could be improved with more depth on the functional role of cannabinoids and the use of cannabinoids in equids.

Response: We added some details of the research carried out in horses by using cannabinoid molecules (lines 53-57).

“Although there is a paucity of university-led research on cannabis therapy in equines, in the last years some interesting studies showed beneficial therapeutic effect of cannabinoid molecules in horses with allodynia[9] degenerative painful conditions such as osteoarthritis and laminitis[10,11], and also behavioural disturbance[12].”

  1. The materials and methods need considerably more detail before they can be suitably evaluated. It is not clear where in each ganglia staining took place or how many slices were obtained. Each ganglion was apparently used for single and double staining, which would suggest different sections/slides were used for each. This should be explained. Was the order standardized or randomized?

Response: In the text, we clarified the sampling method and how the ganglia were sampled (5 for immunofluorescence and 5 for immunohistochemistry, FFPE, from the same animals). The immunofluorescence and FFPE procedures are standardized, the order is random. Other technical details have been added in M&M (lines 357, 371-373; 409-412, 440-443 and Table 3). This issue has also been revised to answer the next question.

  1. Based on the description it reads as if primary ABs from the same host were stained for at the same time in the multiple tag steps. How were these differentiated when they were subsequently stained by the same secondaries? The specific staining rounds are confusing and may benefit from a table. That may allow an opportunity to also illustrate where in each ganglion each staining took place

Response: With the aim to better explain the procedure, some sentences have been changed (lines 371-373). In addition, as suggested, a new table (Table 3) has been added.

  1. The methods surrounding cell counting should also be better explained.  It is not clear why the numbers differ between proteins (e.g. 517, 556, 535, 435, 459, etc.). From the current description, it reads as if this represents 100+ cells from each animal (why that number is not 500 is not clear). Treating each as a separate value is a statistical error. Each animal is an n=1, therefore the average cell number (e.g. out of 100) positive for a protein should be reported as a percentage for each animal (82/100, 75/100, 90/100, etc.) rather than grouped as independent events out of ~500.

Response: In the text we wrote: “A minimum of one hundred Nissl-stained neurons were counted for each cannabinoid and cannabinoid-related receptor in cryosections and paraffin-embedded sections”

We understand the perplexity of the Reviewer. However, we already published a number of articles [see, for instance, Russo et al., 2011 Neuroscience, 176: 53-7; Giancola et al., 2017, Neurogastroenterol Motil Jul;29(7); Chiocchetti et al., 2021 Equine Vet J, 53(3):549-557] in which we counted a number of neurons, which could not be a number lower than one hundred, but which could however be higher, and no Reviewer or Editor ever pointed out to us that it could have been an error; otherwise we would not have counted more than one hundred neurons (per marker and per animal) in the present study. Moreover, in this study, we did not compare different group of immunolabeled neurons or two different group of animals.

  1. The methods indicate that the FFPE was conducted in 5 ganglia however the results have an n=4. Please clarify the discrepancy.

Response: In the Results, lines 135-136, we wrote: ”One paraffin-embedded sample was not adequate for analysis due to an insufficient number of neurons; therefore, four ganglia were used for this study.” However, we highlighted this discrepancy by writing a new sentence (lines 409-412).

  1. The study involved a very wide range of ages (10 months – 22 years) and animal sizes (67kg -374 kg) with such disparity in the sample size, the authors should comment on any trends or differences based on age and size in the results. In addition, a comment should be added to further express the degree of animal variation as an additional study limitation.  
    Response: We agree with the Reviewer and added a new sentence in the discussion (lines 334-338). “There are some limitations which should be taken into consideration when interpreting the results of this study, which considered TG belonging to horses with a very wide range of ages (from 10 months to 22 years) and sizes (from 67 kg to 374 kg). As reported above, the neurons of the older subjects appeared to express fainter immunofluorescence of the studied receptors; however, due to the low number of horses utilized in the current study, any quantitative evaluation were precluded.”

  1. Intext references are needed to several statements without support inclusive of the 1stline of 4.5 where wb analysis in horses is mentioned.

Response: We apologize to the Reviewer because we honestly do not understand his/her request. In fact, in the chapter relating to the specificity of primary antibodies, we have included numerous references relating to studies previously performed (mostly by the present research group) to test the specificity of the antibodies used in the present study.

  1. Also how/where the homology of proteins was determined.
    Response: The source of homology has been added (line 451)

  1. References are also needed for claims in the discussion, for example in the 4thline where trigeminal mediated headshaking is mentioned as the most common neuropathic pain facial condition of horses. In general, statements that cannot be substantiated should be omitted or revised.

Response: A reference has been added, accordingly (line 198). “Ross SE, Murray JK, Roberts VLH (2018) ‘Prevalence of headshaking within the equine population in the UK’ Equine Vet J. Jan;50(1):73-7.”

  1. It is unclear why contrast and brightness adjustments were needed. Adjusting these parameters can result in misinterpreting autofluorescence.
    Response: In M&M we wrote the following sentence: “Slight contrast and brightness adjustments were made using Corel Photo Paint whereas the figure panels were prepared using Corel Draw (Mountain View).”

As is often customary in research using the immunofluorescence technique, this simple procedure only serves to make the images clearer, more readable for the reader, and certainly not to create artefacts or deceive the Reviewers, the Editor and even worse, the Readers. The counting of labelled neurons was conducted by expert researchers, with years of experience, who are able to distinguish between non-existent, weak and effective labelling. Proof of this is the fact that we did not count the neurons labelled by GPR55 in the cryosections, because the immunofluorescence was often too weak or absent to provide reliable numerical values.

  1. Was Nissl staining included as part of the secondary validation (section 4.6)? It reads as if no cells were identified when in fact they should still be identified even if receptors were not tagged.  
    Response: In the present study, the blue fluorescent Nissl stain solution (NeuroTrace®, # N-21479, dilution 1:200) (Molecular Probes, Eugene, OR) has been utilized to label the TG neurons in procedures in which the primary and secondary antibodies (or only the secondary antibodies) have been applied.

However, we understand the perplexity of the Reviewer and changed the sentence (line 463): “The specificity of the secondary antibodies was tested by applying them to the sections after omitting the primary antibodies. No immunolabelledstained cells were detected after omitting the primary antibodies”

  1. Please increase the font size of Figure 5 or break it up into 2 legible figures.
    Response: We modified the figure 5 by increasing the font size, as suggested.

  1. Unfortunately, references can reflect on the quality of the science. Even the use of software should be double-checked before submitting to a journal. Many inconsistencies exist in how references are cited including the use of capital letters and abbreviations for journals. For examples see refs 11, 14, 43. Please double-check references for consistency, accuracy, and journal compliance before submitting.
    Response: We apologize for these mistakes. The references list has been revised accurately.

Reviewer 2 Report

Comments and Suggestions for Authors

In the manuscript titled ‘Expression of cannabinoid receptors in the trigeminal ganglion of the horse. A potential target to treat headshaking’, The authors found that there was expression of cannabinoid receptors in the trigeminal ganglion of the horse. The results are interesting. However, few concerns need to be addressed.

1, The expression of endocannabinoid in trigeminal ganglion in rodents has been well studied. Was mechanism of neuropathic pain in rodents different from that of horses? A discussion is needed to point out the novelty and creative of this research.

2, The ages and body weight of the five animals used in this study have big variation. The authors should report any differences between animals observed in the study.

3, A comparison of the endocannabinoids expression in horses with headshaking would give a better idea that endocannabinoids are underlying the neuropathic pain of horses.

4, There are two endocannabinoids, 2-AG and AEA. Was there any AEA expression? How about expression levels of the enzymes to synthesize endocannabinoids DAGL and NAPE-PLD and the degradation enzymes MAGL and FAAH?   

5, A detailed quantitation of endocannabinoids expression is needed, like the observed cell numbers, the relative expression intensive of each endocannabinoids.

Author Response

Reviewer 2

In the manuscript titled ‘Expression of cannabinoid receptors in the trigeminal ganglion of the horse. A potential target to treat headshaking’, The authors found that there was expression of cannabinoid receptors in the trigeminal ganglion of the horse. The results are interesting. However, few concerns need to be addressed.

Response: We thank the Reviewer for his/her appreciation of the study.

1, The expression of endocannabinoid in trigeminal ganglion in rodents has been well studied. Was mechanism of neuropathic pain in rodents different from that of horses? A discussion is needed to point out the novelty and creative of this research.

Response: The Reviewer is right. There is a number of articles related to the expression of endocannabinoids and cannabinoid receptors in the rodents, and on the therapeutic efficacy of CBD in modulating nociceptive transmission and head pain. However, the study of endocannabinoid system in horses, expecially in horses with headshaking, is in its infancy, and, to the author’s best knowldege, the present study is the first investigation on the cannabinoid receptors within the TG of the horses. One of the reason explaining the paucity of studies on trigeminal ganglion (also just histological studies) of the horse is, perhaps, the scarce availability of equine nervous tissue samples and the complications related to the TG  collection from the horse's head.

Some sentences have been added in the discussion (lines 211-220).

2, The ages and body weight of the five animals used in this study have big variation. The authors should report any differences between animals observed in the study.

Response: We agree with the Reviewer. In fact, we added a sentence related to the limitations of the study (lines: 334-337):

“There are some limitations which should be taken into consideration when interpreting the results of this study, which considered TG belonging to horses with a very wide range of ages (from 10 months to 22 years) and sizes (from 67 kg to 374 kg). As reported above, the neurons of the older subjects appeared to express fainter immunofluorescence of the studied receptors; however, due to the low number of horses utilized in the current study, any quantitative evaluation were precluded.”

3, A comparison of the endocannabinoids expression in horses with headshaking would give a better idea that endocannabinoids are underlying the neuropathic pain of horses.

Response: We totally agree with Reviewer. In fact we are organizing the next step of the research, by collecting (FFPE) tissues from horses with trigeminal mediated headshaking which have been collected in some UE and extra-UE countries. Thius is a time-consuming procedure, and this project doesn't fit with the deadline of the specific call on cannabinoid receptors. However, due to the fact that in horses no studies are available on the distribution of cannabinoid receptors in the TG, we thought to present this first basic study.

4, There are two endocannabinoids, 2-AG and AEA. Was there any AEA expression? How about expression levels of the enzymes to synthesize endocannabinoids DAGL and NAPE-PLD and the degradation enzymes MAGL and FAAH?  

Response: We thank the Reviewer for this comment. The evaluation of the endocannabinoids, such as the enzymes for the synthesis and degradation of endcannabinoids will be the topic of the next studies on the TG of the horses. Before this next investigation, however, we must test the specificity of the antibodies on horses; this is not a simple procedure, because all the commenrcally available antibodies are specific for human and rodents, whereas no anntibodies have been tested in large animals such as horses.

5, A detailed quantitation of endocannabinoids expression is needed, like the observed cell numbers, the relative expression intensive of each endocannabinoids.

Response: We understand and appreciate the comment of the Reviewer; however, the present study is specific on the expression of cannabinoid and cannabinoid-related receptors. We plan to conduct further studies in which the endocannabinoids will be elaluated and studied. In the current study we evaluated the percentage of TG neurons expressing the studied receptors and prepared a table in which the receptors were semiquantitatively evaluated.

Reviewer 3 Report

Comments and Suggestions for Authors

I have been extremely impressed by this manuscript and would like to commend the authors on some good quality research, with beautiful illustrations.

I have only a few, very minor, comments.

One general comment – the symbol used for gamma seems to differ when used in different places in the manuscript.

Results

Section 2.1

The authors mention ‘immunofluorescence expressed by the cytoplasm… or nerve fibres’, in three places. I wonder if it would be less confusing to say, ‘immunofluorescence expressed by the cytoplasm of nerve cell bodies… or nerve fibres’.

Paragraph starting, ‘Peroxisome proliferator-activated receptor gamma immunofluorescence’… on the fourth line, you state: ‘in some cells in the nuclei were moderately reactive’. I think this would be better without the second ‘in’, e.g., ‘in some cells, the nuclei were moderately reactive’.

Figure 2. Legend for (a-f), ‘The open arrow indicates two TRPV1…’ would be better as, ‘The open arrows indicate two TRPV1...’.

Section 2.2

Again, ‘expressed by the cytoplasm’ and ‘cytoplasmic’, appear four times. Do the authors mean ‘cytoplasm of the nerve cell bodies’?

Figure 4. Legend for (f), ‘The open arrows indicate’ should, I think, be, ‘The open arrow indicates’ (as I can only see one arrow).

Discussion

Page 9, Para 2: The authors introduce a new acronym (HSK), presumably meaning ‘head-shaking’, but in the Introduction the authors use the term ‘TGM headshaking’ so I wasn’t sure if the authors wanted to be consistent with the terminology used in the Introduction.

Materials and Methods

Section 4.1 (Page 12, first line): ‘median plane of the head’. Do the authors mean the sagittal plane?

Section 4.2 (Page 12, para 2): ‘RT’ should be written in full (room temperature) when first used (unless I’ve missed this?).

Section 4.4 (Page 14, first two lines): ‘cellular localisation (membranous, cytoplasmic and/or nuclear)’. Again, I wonder whether to make the distinction that when the term ‘cytoplasmic’ is used, it means nerve cell body cytoplasm??

Conclusions

Page 15, Line 2: please insert the word ‘and’ before, ‘cannabinoid-related receptors’.

Last two lines of Conclusion: ‘thereby enhancing the well-being, performance and quality of life’ should be expanded to, ‘thereby enhancing the well-being, performance and quality of life of these horses’.

Author Response

Reviewer 3

I have been extremely impressed by this manuscript and would like to commend the authors on some good quality research, with beautiful illustrations. I have only a few, very minor, comments.

Response: We thank the Reviewer for his/her appreciation of the study.

1 - One general comment – the symbol used for gamma seems to differ when used in different places in the manuscript.

 Response: We have checked throughout the text and it seems that the gamma symbol is uniform (Arial 10).

Results

Section 2.1

2- The authors mention ‘immunofluorescence expressed by the cytoplasm… or nerve fibres’, in three places. I wonder if it would be less confusing to say, ‘immunofluorescence expressed by the cytoplasm of nerve cell bodies… or nerve fibres’.

Response: We thank the Reviewer for his/her comment. It is important to localize the receptor, which may be expressed by the cell membrane, cytoplasm, nucleus, nucleolus and neuronal process/fibres of the neurons. “Cytoplasm” has been changed with “cell body cytoplasm” throughout the text.

3- Paragraph starting, ‘Peroxisome proliferator-activated receptor gamma immunofluorescence’… on the fourth line, you state: ‘in some cells in the nuclei were moderately reactive’. I think this would be better without the second ‘in’, e.g., ‘in some cells, the nuclei were moderately reactive’.

 Response: We modified the sentence, accordingly.

4 - Figure 2. Legend for (a-f), ‘The open arrow indicates two TRPV1…’ would be better as, ‘The open arrows indicate two TRPV1...’.

 Response: We thank the Reviewer for the accuracy of his/her revision; we modified the sentence, accordingly.

Section 2.2

5- Again, ‘expressed by the cytoplasm’ and ‘cytoplasmic’, appear four times. Do the authors mean ‘cytoplasm of the nerve cell bodies’?

Response: As specified above, “cytoplasm” (or “cytoplasmic”) has been changed with “cell body cytoplasm” (of the neurons) throughout the text.

6 - Figure 4. Legend for (f), ‘The open arrows indicate’ should, I think, be, ‘The open arrow indicates’ (as I can only see one arrow).

 Response: We modified the sentence, accordingly.

Discussion

7- Page 9, Para 2: The authors introduce a new acronym (HSK), presumably meaning ‘head-shaking’, but in the Introduction the authors use the term ‘TGM headshaking’ so I wasn’t sure if the authors wanted to be consistent with the terminology used in the Introduction.

 Response: Thank you very much for this correction. We modified the acronym, by adding “TGM headshaking”.

Materials and Methods

8- Section 4.1 (Page 12, first line): ‘median plane of the head’. Do the authors mean the sagittal plane?
Response: The Reviewer is right, we meant “sagittal plane”; the word has been changed.

9 - Section 4.2 (Page 12, para 2): ‘RT’ should be written in full (room temperature) when first used (unless I’ve missed this?).
Response: Correct, the “room temperature” was missing in the text; the text has been changed, accordingly.

10 - Section 4.4 (Page 14, first two lines): ‘cellular localisation (membranous, cytoplasmic and/or nuclear)’. Again, I wonder whether to make the distinction that when the term ‘cytoplasmic’ is used, it means nerve cell body cytoplasm??
Response: Yes, with “cytoplasmic” we indicate the cytoplasm of the nerve body. We changed “cytoplasmic” with “cell body cytoplasm” throughout the text.

Conclusions

11- Page 15, Line 2: please insert the word ‘and’ before, ‘cannabinoid-related receptors’.
Response: Done.

12- Last two lines of Conclusion: ‘thereby enhancing the well-being, performance and quality of life’ should be expanded to, ‘thereby enhancing the well-being, performance and quality of life of these horses’.
Response: We thank the Reviewer for your kind corrections. The sentence has been modified as suggested.

Round 2

Reviewer 1 Report

Comments and Suggestions for Authors

I would like to thank the authors for making changes in the manuscript in response to all reviewer feedback that improved the quality of the paper.

The author did not provide justification for why the cell counts differed between animals and could not be 100 each. I assume there is a reasonable explanation, but this should be included. 

I appreciate the addition of Table 3. It is still unclear how many slices were stained with each protocol method. This should be added to Table 3. 

Although the selection of location for each technique was reportedly random. The results could represent regional differences. That information if not included needs to be added as a study limitation before acceptance. 

The authors indicated that they reviewed and corrected inconsistencies in the references. These were not changed. For example see references 15, 19, 66, 37, 75. These blaring inconsistencies are evident with just a cursory scan. Do not claim to make corrections if you did not do so. 

The authors should also note that many of the initial reviewer comments from me and my colleagues involved simple errors that should have been caught through proofreading before submitting this paper to your peers to evaluate.  

Author Response

I would like to thank the authors for making changes in the manuscript in response to all reviewer feedback that improved the quality of the paper.

We are very grateful to the Reviewer for giving us an additional opportunity to clarify and specify some points in our paper, clarifying some doubts and correcting some inaccuracies that we had missed or had taken for granted at the drafting stage of the paper because they were routine for us. We then tried to respond more thoroughly to doubts about some aspects of our paper and to edit the text accordingly when necessary.

The author did not provide justification for why the cell counts differed between animals and could not be 100 each. I assume there is a reasonable explanation, but this should be included.

Answer: the counting procedures for IHC are different between immunofluorescence and FFPE techniques, and are standardized for each method. In the M&M of the paper (paragraph 4.4 Quantitative analysis of the immunoreactivity), we have tried to make this as clear as possible, and we apologize for the fact that perhaps, because we are used to proceeding in this way and because in much of our work this procedural aspect is taken for granted, we had not initially thought on detailing it in depth.

This type of counting is done for immunofluorescence evaluation: cells immunostained for cannabinoid receptors were counted directly by observing the slides under a fluorescence microscope. The technique involves groups of cells stained with Neurotrace (fluorescent Nissl staining) and, in each field, we dictate to an assistant (who records the number on a sheet) the number of immunolabeled neurons. When the assistant confirms that we have recorded at least 100 Neurotrace-labeled neurons, we stop counting, but never by field count. It is then verified that the number of neurons counted can also be more than 100, because the field count always has to be concluded.

For FFPE, on the other hand, all cells in the same evaluated compartment (neurons) in 5 fields on the slide were counted, evaluated at 20x magnification with the light microscope, and then the % of immunopositive cells in the total was calculated and then averaged among the cases.

I appreciate the addition of Table 3. It is still unclear how many slices were stained with each protocol method. This should be added to Table 3.

Answer: for each protocol method, one slice was made for each antibody evaluated and, of course, this multiplied by the five (immunofluorescence) and four (FFPE) sample cases (TG). For FFPE slices in total (20 sections), n=20 slides were evaluated. For immunofluorescence slices in total (62 cryosections), n=22 slides were evaluated.

The required specifications have been added to the text in the M&M section (Line 404-405 and 458).

Although the selection of location for each technique was reportedly random. The results could represent regional differences. That information if not included needs to be added as a study limitation before acceptance.

Answer: we did not fully understand whether by location you actually mean the anatomical location of the sampling or the selection of the fields in the slides for the observation with both methods. The selection of the total evaluable area for each technique was not random and was the same for both, and the site from which the portion of the tissue sample was taken was the same (right or left TG) of the same horse. Of course, in some cases it might not have included the ganglion in full, since the whole ganglion could not have been contained in the processing cages (immunohistochemistry) or cryomolds (immunofluorescence). In the processed sections, however, there was always an extensive portion of ganglion and an adjacent portion of nerve fibres present close to the emergence of the three branches (ophthalmic, maxillary and mandibular). Indeed, the sampling had included about 1,5 cm of trigeminal nerve, rostral to the emergence of the three branches. We have however modified in the discussion the limitations of our paper by specifying them better (line 365-368).

The authors indicated that they reviewed and corrected inconsistencies in the references. These were not changed. For example see references 15, 19, 66, 37, 75. These blaring inconsistencies are evident with just a cursory scan. Do not claim to make corrections if you did not do so.

Answer: We apologize again very much for possible oversights. We thank you very much for your patience and painstaking revision work.

The authors should also note that many of the initial reviewer comments from me and my colleagues involved simple errors that should have been caught through proofreading before submitting this paper to your peers to evaluate. 

Answer: We apologize again very much for possible oversights or superficialities in the first version of the work, dictated in part by superficial considerations regarding the unnecessary detailing of already standardized procedures, and the lack of clarity in some aspects. We thank you again for your patience and painstaking correction work.

Reviewer 2 Report

Comments and Suggestions for Authors

The authors should have put more emphasis on addressing the reviewers' concerns. 

1, With the big variation on age and body weight of the horses used in this study, it was not ready to reach the conclusions, especially when the older subjects expressed fainter endocannabinoids receptors was observed in the study.  

2, Without a comparison of the endocannabinoids expression between normal and headshaking horses, it was too early to state "A potential target to treat headshaking".

Author Response

The authors should have put more emphasis on addressing the reviewers' concerns.

We apologize very much to the Reviewer for not explaining the reasons for our answers more precisely and in more detail.

1, With the big variation on age and body weight of the horses used in this study, it was not ready to reach the conclusions, especially when the older subjects expressed fainter endocannabinoids receptors was observed in the study. 

Answer: our speculations did not have the character of conclusions, but we expressed a hypothesis based of our observations, albeit with limitations (in the text, line 82-83 and 365-368) regarding non-homogeneous age and weight, as well as the paucity of cases, which did not allow for statistically significant considerations in this regard. The reasons for this observation derive from the fact that the lower expression of receptors in elderly subjects may be an indication of the depletion of the endocannabinoid system in old age, and this could be useful information to support ECS in elderly animals, perhaps by administering endocannabinoids or molecules that can enhance endocannabinoid metabolism.

2, Without a comparison of the endocannabinoids expression between normal and headshaking horses, it was too early to state "A potential target to treat headshaking".

Answer: before studying the expression of endocannabinoids and their potential role in pathology, it becomes necessary to evaluate the expression of cannabinoid receptors in healthy control tissue, something that in the TG of the horse has never been done (hence the reason why we focused on this first part of investigation). The pathology targets the trigeminal ganglion and, above all, is a pathology that has neuropathic basis but histologically has no different features from what tissue under normal conditions might have. However, we understand the perplexity of the Reviewer and changed the title.